# Comprehensive Investigation of Die-Back Disease Caused by *Fusarium* in Durian

**DOI:** 10.3390/plants12173045

**Published:** 2023-08-24

**Authors:** Ratiya Pongpisutta, Pisut Keawmanee, Sunisa Sanguansub, Paradorn Dokchan, Santiti Bincader, Vipaporn Phuntumart, Chainarong Rattanakreetakul

**Affiliations:** 1Department of Plant Pathology, Faculty of Agriculture at Kamphaeng Saen, Kasetsart University, Nakhon Pathom 73140, Thailand; ratiya.p@ku.th (R.P.); pisut.keaw@ku.th (P.K.); 2Department of Entomology, Faculty of Agriculture at Kamphaeng Saen, Kasetsart University, Nakhon Pathom 73140, Thailand; agrssss@ku.ac.th (S.S.); paradorn.d@ku.th (P.D.); 3Program Plant Science, Faculty of Agricultural Technology and Agro-Industry, Rajamangala University of Technology Suvarnabhumi, Phra Nakhon Si Ayutthaya 13000, Thailand; santiti.b@rmutsb.ac.th; 4Department of Biological Sciences, 129 Life Sciences Building, Bowling Green State University, Bowling Green, OH 43403, USA; vphuntu@bgsu.edu

**Keywords:** *Fusarium*, durian, pathogenicity, ambrosia beetle, transmission

## Abstract

Durian (*Durio zibethinus* L.) is an economically important crop in the southern and eastern parts of Thailand. The occurrence of die-back disease caused by plant pathogenic fungi poses a serious threat to the quality and quantity of durian products. However, the identification of causal agents has been a subject of mixed information and uncertainty. In this research, we conducted a comprehensive investigation of die-back disease in nine durian plantations located in Thailand. By analyzing a total of 86 *Fusarium* isolates obtained from infected tissues, we aimed to provide clarity and a better understanding of the fungal pathogens responsible for this economically significant disease. Through a combination of colony characteristics, microscopic morphology, and a multilocus sequence analysis (MLSA) of the internal transcribed spacer (ITS) region, translation elongation factor 1-α (*TEF1*-*α*) gene, and RNA polymerase II gene (*RPB2*) sequences, we were able to identify and categorize the isolates into three distinct groups, namely, *Fusarium incarnatum*, *F. solani*, and *F. mangiferae.* Koch’s postulates demonstrated that only *F. incarnatum* and *F. solani* were capable of causing die-back symptoms. This research represents the first report of *F. incarnatum* as a causal agent of die-back disease in durian in Thailand. Additionally, this study uncovers the association of ambrosia beetles and *F. solani*, highlighting the potential involvement of *E. similia* in facilitating the spread of die-back disease caused by *Fusarium* in durian.

## 1. Introduction

Durian (*Durio zibethinus* L.) holds significant economic importance as a fruit crop in various Southeast Asian countries, including Viet Nam, Indonesia, Malaysia, and notably Thailand which leads the global durian export market [1]. The increasing demand for durians in China has stimulated continuous growth in Thailand’s durian export and driven up durian prices in recent years. In 2022, China’s export volume reached an impressive THB3.14 billion, representing a remarkable year-on-year growth of 108.08 percent, thereby contributing to the surge in durian prices [2]. Moreover, the data from Chinese customs indicate a substantial increase in both the import value and volume of China’s fresh durians, reaching THB4.21 billion in the previous year. This represents an impressive 82.4 percent growth in value compared to 2020 [3]. 

In recent years, there has been a significant expansion of durian plantation areas, particularly in the southern and eastern parts of Thailand, with a total of more than 1,099,228 rai being cultivated [4]. While the cultivation of durian has experienced significant growth, it has also been affected by various plant diseases, such as root and collar rot, caused by *Phytophthora palmivora* [5]. The diseases have been a leading cause of crop losses of around 20–25% in previous years [5,6,7]. In 2018–2019, Thai durian growers faced the issue of a die-back disease outbreak, and the predominant suspect for the disease was *P. palmivora* [5,7]. Despite growers’ attempts to control the disease by applying fungicides to target *P. palmivora*, these efforts proved ineffective in controlling durian die-back. To address this issue, a survey was conducted in the eastern and southern regions of Thailand, aiming to investigate and collect samples of durian die-back disease [8]. The disease’s symptoms included yellow leaves, which eventually dropped from the branches, and the branch tips manifested grooves of the bark. This durian die-back outbreak rapidly spread across many plantation areas where the growers faced challenges due to a lack of manpower and resources for intensive investigation. We recognized the importance of accurately identifying the causative pathogens at the initial stage of the investigation.

Several species of ambrosia beetles are known for their unique behavior of cultivating and feeding on specialized fungi. In return, the beetles provide the fungi with a protected and nutrient-rich environment for their growth and reproduction [9,10,11,12]. *Euwallacea similis* is a species of ambrosia beetle native to Asia, including Thailand [13,14]. They are typically found in stressed and dead trees. The beetle introduces the pathogenic fungus into the tree’s vascular system while excavating its galleries, leading to the clogging of the xylem vessels and the wilting of the tree. The association of *E. similis* with fungi, which serve as their food throughout all life stages, is carried by the adult female and can be introduced to any new habitat [15]. One notable example of this symbiosis is the transmission of *Fusarium* spp. by the ambrosia beetle *Euwallacea* sp. This relationship has significant implications in the spread of fungal wilt disease in *Acacia crassicarpa* [16]. By studying the interactions between *E. similis* and *Fusarium* in the context of durian plantations, researchers can shed light on the role of ambrosia beetles in disease transmission and devise measures to mitigate potential losses caused by *Fusarium*-related diseases. 

Therefore, this research’s primary objectives were to identify the specific pathogens responsible for causing die-back, to conduct pathogenicity assays on durian seedlings to confirm the pathogens’ capability to induce the disease, and to investigate the role of the ambrosia beetle in spreading the durian die-back disease. The significance of this study is that, firstly, this research represents the first report of *Fusarium incarnatum* as a causal agent of die-back disease in durian in Thailand. Prior to this study, there was limited information available on the specific fungal pathogens responsible for this economically important disease in durian plantations. By identifying *F. incarnatum* as one of the causative agents, we have expanded the knowledge base and added to the understanding of the complex interactions between the pathogen and the host plant. Secondly, this present research is the first report of durian die-back disease and its association with the ambrosia beetle in Thailand. The mutualistic association between the ambrosia beetle and *Fusarium* highlights the significant role of the ambrosia beetle in the transmission and spread of the disease-causing fungus. This finding underscores the importance of considering both the fungal pathogen and its associated insect vectors when formulating effective disease management strategies in durian plantations. 

## 2. Results

### 2.1. Fungal Isolation and Purification

In this study, a total of 86 fungal isolates resembling *Fusarium* spp. were isolated from infectious samples collected from durian plantations located in three different provinces: Chumphon (CHU), Chanthaburi (CHA), and Trat (TRA) (Figure 1).

The occurrence and prevalence of die-back diseases vary from season to season, depending on weather conditions and the management condition of each plantation. In the regions investigated, the period from April to July corresponds to the monsoon season and is warm (26–31 °C), where the incidence of the disease can reach 70–80% of the total plant. The primary symptom observed included rapid leaf fall, typically occurring within 5 to 7 days of infection. Following the leaf drop, the infected branches exhibited die-back symptoms, where the affected parts of the tree began to wither and die. Upon close examination of the infected twigs, brownish to dark brown lesions were evident, indicating the presence of the disease (Figure 2). These findings indicate that the fungal isolates resembling *Fusarium* spp. could be associated with the die-back disease.

### 2.2. DNA Marker and Fungal Identification

To expedite the identification process of the isolated fungal species, a total of 86 isolates resembling *Fusarium* were separated into 17 groups based on colony characteristics. One isolate from each group was then selected for a DNA marker analysis. The DNA sequences of ITS, *TEF1-α*, and *RPB2* were analyzed using BLASTn to compare them with known sequences of *Fusarium* species in a public database, available at the GenBank National Center for Biotechnology Information (NCBI). For the phylogenetic analysis, initially, a phylogenetic tree was constructed using only the ITS1-5.8s-ITS2 region. Later, the analysis was expanded to include the *TEF1-α* and *RPB2* markers to obtain more comprehensive information on the genetic relationships among the isolates. This involved constructing a concatenated phylogenetic tree using the combined sequences of ITS, *TEF1-α*, and *RPB2*.

For a phylogenetic tree using only the ITS1-5.8s-ITS2 region, a total of 17 isolates, including 20 isolates of ex-type and ex-epitype or epitype strains [17,18,19,20,21,22], along with *Fusarium staphyleae* NRRL22316 [17] as an outgroup, were utilized (Table 1). The resulting phylogenetic tree of *Fusarium* isolates, based on the maximum parsimony (MP) analysis of the ITS region, is shown in Figure 3.

According to the analysis, the 17 isolates were classified into three main groups: *F. solani*, *F. mangiferae*, and *F. incarnatum-equiseti* species complex (Figure 3). Among these, 11 isolates (DCHU403, DCHU 1404, DCHU801, DCHA1708, DCHU107, DCHU303, DCHA904, DCHA902, DCHA408, DCHA102, and DCHA424) formed a clade within the *F. solani* species complex. They were closely related to the *F. solani* strains (NRRL32810, NRRL32791, and NRRL32741) and the *F. solani* f.sp. *batatas* strain (NRRL22400), showing 0.00 posterior probability (PP) and 99% bootstrap (BS). One isolate (DCHA805) was grouped with *F. mangiferae* (CBS119853) with 0.03 posterior probability and 99% bootstrap. The remaining five isolates (DCHA1301, DCHA1101, DCHA1607, DCHA704, and DCHA1801) clustered within the *F. incarnatum-equiseti* species complex with *F. incarnatum* (NRRL32866, NRRL32867, and NRRL13379), *F. camptoceras* (CBS193.65), *F. brevicaudatum* (NRRL43638), *F. arcuatisporum* (NRRL32997), *F. compactum* (NRRL28029), and *F. equiseti* (NRRL36136) with 0.00 posterior probability and 98% bootstrap. The results suggested a high confidence level in the genetic relationship between these isolates with their related species. Intriguingly, an analysis of the distribution and relative abundance of *Fusarium* across the selected sites and sampling points revealed that *F. incarnatum* was the predominant species (61.63%), followed by *F. solani* (34.88%) and *F. mangiferae* (3.49%), as documented in Appendix A.

For a concatenated phylogenetic tree using three markers, ITS, *TEF1-α* and *RPB2*, a total of 38 isolates were analyzed, including 20 isolates of ex-type and ex-epitype or epitype strains, along with *F. staphyleae* NRRL22316 as an outgroup (Figure 4). The results showed that eleven isolates clustered with the *F. solani* strains (NRRL32741, NRRL32810, and NRRL32791), showing 0.01 posterior probability and 97% bootstrap. One isolate formed a cluster with the *F. mangiferae* strain (CBS119853) with 0.00 posterior probability and 100% bootstrap. Of particular importance, the remaining five isolates, including DCHA704, clustered with the *F. incarnatum* strains (NRRL32867 and NRRL32866) with a posterior probability of 0.00 and a bootstrap value of 99%. Notably, DCHA704 was completely separated from clustering with the *F. incarnatum-equiseti* species complex, particularly from the *F. equiseti* strain (NRRL36136). These results indicate that DCHA704 is distinct from *F. equiseti* and is more closely related to *F. incarnatum*.

Overall, the concatenated phylogenetic tree provided better resolution and valuable insights into the genetic relationships and relatedness of the analyzed *Fusarium* isolates. The clustering patterns indicated the presence of specific species groups, such as *F. solani*, *F. mangiferae*, and *F. incarnatum*, among the durian die-back samples collected from the eastern and southern regions of Thailand.

### 2.3. Morphological Characteristics

To complement the phylogenetic analysis, we conducted examinations of the colony characteristics and microscopic morphology. All the 86 isolates were assessed for colony characteristics, and representative isolates from the three groups were further examined. The first group, identified as *F. solani*, included isolates DCHA408 (gr.4), DCHU303 (gr.5), DCHU107 (gr.7), DCHA902 (gr.8), DCHU801 (gr.9), DCHA1708 (gr.10), DCHU403 (gr.13), DCHU1404 (gr.14), DCHA102 (gr.15), DCHA904 (gr.16), and DCHA424 (gr.17). These isolates exhibited variations in color, ranging from white or cream to dark violet, and produced fluffy mycelia. Some isolates displayed pigments in the agar, with the color ranging from brown to violet when viewed from the reverse side of the agar plate (Figure 5A). Generally, the colony of each isolate contained a higher number of macroconidia compared to microconidia (Figure 5B). The macroconidia were straight, wide, and thick-walled, with a blunt and rounded apical cell and a distinct foot shape with a notched or rounded end in the basal cell. The length of the macroconidia ranged from 22.50 to 55.00 µm, and the widths ranged from 2.50 to 5.00 µm, with four to six septa. The microconidia were observed to have oval, ellipsoid, and reniform shapes with zero or one septum. They were produced on long conidiophores, and the conidiogenous cells were monophialides with false heads (Figure 5C). The lengths of the microconidia varied from 7.50 to 20.00 µm, and the widths ranged from 1.86 to 3.75 µm. This characteristic is consistent with the typical growth pattern of *Fusarium* species, where macroconidia are the predominant type of spores produced and are often more abundant than microconidia in the colony. The macroconidia are larger and play a significant role in the dispersal of the fungus, contributing to the pathogen’s ability to infect and spread within the host plant and the environment. On the other hand, microconidia are smaller and usually produced in lesser quantities. Furthermore, these isolates abundantly formed chlamydospores in the PDA culture (Figure 5D), and hyphal coils were also observed (Figure 5E).

The second group, identified as *F. incarnatum*, consisted of DCHA704 (gr.1), DCHA1301 (gr.2), DCHA1801 (gr.3), DCHA1607 (gr.6), and DCHA1101 (gr.11). These isolates exhibited distinct morphological characteristics that distinguished them from other *Fusarium* species. They produced abundant dense aerial mycelia that initially appeared off-white and later turned yellowish-brown with age. Brown pigments were also produced in the agar, with some isolates displaying salmon-colored pigment (Figure 6A). The macroconidia were slender with a curved dorsal surface and a straighter ventral surface (Figure 6B,C). The apical cell was curved and tapered to a point. The basal cell exhibited a distinct foot shape. The macroconidia had four to six septa, with lengths ranging from 21.03 to 41.15 µm and widths from 2.92 to 4.07 µm. The microconidia were ovoid in shape with a single cell, measuring from 7.17 to 24.00 µm in length and 2.35 to 4.00 µm in width (Figure 6D). Furthermore, chlamydospores were abundantly formed in these isolates and appeared globose or ellipsoid in shape. The chlamydospores were typically intercalary or terminal, hyaline, and had smooth walls. Their size varied from 5.35 to 17.40 µm in length and 6.10 to 16.07 µm in width (Figure 6E). These distinctive morphological characteristics helped to confirm the identification of the five isolates as *F. incarnatum*, providing valuable information for understanding their potential role as causal agents of die-back disease in durian plantations.

The third group, identified as *F. mangiferae*, was represented by a single isolate, DCHA805 (gr.12). This isolate displayed white and purple floccose mycelia with light to dark purple pigments on the reverse side of the agar plate (Figure 7A). The macroconidia were thin-walled and slightly curved with three to five septa (Figure 7B). The apical cell was slightly curved, and the basal cell exhibited a foot-shaped structure. The sizes of the macroconidia ranged from 17.53 to 35.08 µm in length and 1.84 to 2.40 µm in width. Abundant microconidia with one-to-two cells, displaying oval and allantoid shapes, were produced on phialides as the false head (Figure 7C). The microconidia sizes were investigated, with the lengths ranging from 3.12 to 10.02 µm and the width ranging from 1.15 to 2.44 µm. Furthermore, chlamydospores were abundantly formed in this isolate, mostly appearing as globose to ellipsoid structures. The chlamydospores were typically intercalary or terminal, hyaline, and had smooth walls. They were solitary and exhibited a size of 4.27 to 14.56 µm in length and 4.33 to 15.66 µm in width (Figure 7D). The unique morphological features of this isolate helped confirm its identification as *F. mangiferae*.

### 2.4. Pathogenicity Test

This study conducted pathogenicity assays by inoculating seventeen representative *Fusarium* isolates on durian seedlings’ stems. The results showed that both *F. incarnatum* and *F. solani* were identified as pathogenic fungi, causing disease on the seedlings. In contrast, *F. mangiferae* was found to be non-pathogenic. The disease symptoms started to develop seven days after inoculation, and the average lesion length recorded was 2.78 cm for *F. incarnatum* and 4.48 cm for *F. solani*, with a significant difference (*p* < 0.05). In comparison, the control group (not inoculated with the fungi) did not show any lesion formation (Table 2; Figure 8). The observed necrosis symptom on the bark tissue appeared brown, and the lesion extended into the wood, leading to the cell death of the inoculated seedling within 21 days after inoculation (Figure 8).

The statistical analysis further confirmed the significant differences in the lesion length between the control group and the young stems inoculated with the fungi. Following the pathogenicity assay, all the *Fusarium* pathogenic species were successfully re-isolated from the infected tissues, and their morphological characteristics matched the prototype isolates, fulfilling Koch’s postulates. This process validated their role as the causal agents responsible for the observed disease symptoms on the durian seedlings.

### 2.5. Fusarium–Ambrosia Beetle Association

During the disease survey at durian plantations, sawdust was observed along with the die-back symptoms and the presence of *Fusarium* species (Figure 9A–C). To investigate the association between ambrosia beetles and the *F. solani* isolate CHA-TW-NOV-1-3/1, the ambrosia beetles were provided with an artificial medium composed of durian sawdust, with and without *F. solani*. When the ambrosia beetles were fed with the artificial medium containing *F. solani*, they were able to grow in galleries and they bored into the medium (Figure 9D–I). The survival rate of the ambrosia beetles was 80% after 5 days of incubation at room temperature in the dark. However, in the control group where the medium did not contain *F. solani*, no ambrosia beetles survived. Furthermore, the digestive tracts of the ambrosia beetles transferred to the PDA mixed with streptomycin, showing fungal growth with characteristics similar to the original *F. solani* isolate (Figure 9J–L).

This finding further confirms the association between ambrosia beetles and *F. solani*, as the fungus was able to grow and colonize the digestive tracts of the beetles. The ability of the fungus to survive and grow within the digestive tracts of the beetles suggests a mutualistic relationship, where the beetles may benefit from the fungus as a food source or in other ways, while the fungus may benefit from being transported and disseminated by the beetles to new locations.

## 3. Discussion

This study emphasizes the importance of early detection and timely management to mitigate the impact of die-back disease on durian plantations and prevent further spread of the disease. To identify the *Fusarium* species, present in the samples, this study employed multiple DNA markers, including the ITS1-5.8s-ITS2, *TEF1-α*, and *RPB2* regions, to rapidly identify the *Fusarium* species present in the samples. This molecular diagnostic approach significantly sped up the identification process, enabling us to identify and categorize the *Fusarium* species accurately. Similar to other studies, the MLSA demonstrated better resolution at the species level compared to individual gene analyses. For instance, a study by Rattanakreetakul et al. [23] compared two phylogenetic trees for *Colletotrichum* species. The MLSA approach, which used four loci (ITS, *TUB2*, *ACT*, and *CHS-1*), provided more accurate identification, distinguishing them into *C. acutatum*, *C. gloeosporioides*, *C. asianum*, and *C. siamense*, compared to using only two loci (ITS and *TUB2*). Three genetic barcodes, *TEF1-α*, *RPB2*, and *TUB2*, have been recognized as highly informative and effective markers for studying the phylogeny and diversity of *Fusarium* species [24,25,26,27,28]. Therefore, in this study, the MLSA approach combining these three highly informative genetic barcodes was employed. This approach allowed for a more comprehensive and accurate assessment of the relationships between different *Fusarium* isolates, facilitating the identification and classification of specific species within the complex. The MLSA effectively distinguished the *F. incarnatum* strain from the *F. incarnatum-equiseti* species complex, showcasing its efficacy as a powerful tool in fungal taxonomy and systematic studies. It provides a robust and reliable method for distinguishing closely related species and clarifying the diversity and distribution of *Fusarium* species in different regions.

Due to the time-consuming nature of detailed morphological inspections and the challenges in assessing mycelia characteristics, colony colors, pigmentations, conidiogenous cells, and the presence of monophialide or polyphialide, [25,28,29,30], we initially employed MLSA for more rapid identification. Subsequently, colony characteristics and microscopic morphology were conducted to confirm the identity of the isolates. *F. solani* displayed variations in color on agar, ranging from white or cream to dark violet. Some isolates exhibited pigments ranging from brown to violet when viewed from the reverse side of the agar plate. They predominantly produced relatively long monophialides compared to other species and contained a higher number of macroconidia than microconidia. *F. incarnatum* displayed a wide range of colors, initially producing off-white mycelia that later turned yellowish-brown with age. The pigments produced by these isolates ranged from salmon colored to brown. Abundant chlamydospores were formed in these isolates, which appeared globose or ellipsoid in shape. They produced relatively smaller macroconidia compared to *F. solani.* Similarly, *F. mangiferae* also exhibited variations in morphology, displaying white and purple floccose mycelia with light to dark purple pigments on the reverse side of the agar plate. Their macroconidia were the smallest compared to *F. solani* and *F. incarnatum.*

Evidently, relying solely on one approach for identification can be insufficient, especially when dealing with closely related species or cryptic species that may show similar morphological characteristics. In such cases, a combination of MLSA and morphology provides more accurate tools for identification. Additionally, MLSA can determine the phylogenetic relationships and genetic relatedness between the isolates and known species. This approach provides a more robust and reliable method for species identification, especially when dealing with species complexes or closely related taxa. On the other hand, morphological observations are still valuable for the confirmation and accuracy of the classification and characterization of fungal isolates, especially when dealing with fungi that have variable or overlapping morphological characteristics.

Indeed, the results of this study highlight the significance of *F. solani* and *F. incarnatum* as the key causal agents responsible for die-back disease in durian plantations in Thailand. *F. solani* was the most frequently isolated fungus from the diseased tissues, indicating its prevalence and importance as a pathogen. This finding is consistent with a previous report in 2020 [8], which also identified *F. solani* as a pathogenic fungus causing die-back disease in durian. However, this study also brought to light a new discovery, as it revealed, for the first time, that *F. incarnatum* is another crucial causal agent responsible for durian die-back in Thailand. This is an important finding, as it expands our understanding of the diversity of pathogens involved in the disease and emphasizes the need to consider multiple species of *Fusarium* in disease management strategies. Differentiating between the specific pathogens causing the disease is essential for implementing effective control measures and preventing further spread of the disease.

The *Euwallacea* beetles–*Fusarium* fungi symbiosis has been documented in several countries and regions, including Israel [31], the United States [17,32,33], Panama, Costa Rica [34], and Taiwan [35]. In some cases, this association has led to severe diseases in tree species, impacting forest health and urban landscapes [36,37]. Our study demonstrated the survival of ambrosia beetles on artificial ambrosia medium inoculated with *F. solani*, suggesting a potential symbiotic relationship between the ambrosia beetles and the *Fusarium* species. The presence of *F. solani* inside the digestive tract of the ambrosia beetles provided additional evidence supporting the idea of a close interaction between ambrosia beetles and *Fusarium*. The success of ambrosia beetles as invaders has been reported in several studies [38,39,40,41] and can be attributed, in part, to their wide host range and their association with the primary ambrosia fungi, such as *Fusarium* species, which plays a crucial role in their survival and colonization in new environments. The fungi in this relationship may act as both nutritional symbionts and weak phytopathogens [21,32]. There are significant variations in the fungal symbionts, geographic range, host preference, and potential for symbiont switching in natural populations of these beetles [42,43].

The findings contribute to a deeper understanding of the complex relationships between insects and their associated fungi, shedding light on their role as successful invaders and potential agents of plant disease in different ecosystems. Particularly, this study highlights the potential role of *E. similis* in spreading die-back disease in durian trees in Thailand. Overall, this study’s findings contribute significantly to the knowledge of durian die-back disease and underscore the importance of continuous research and monitoring to stay ahead of the challenges posed by plant diseases in agricultural settings.

## 4. Materials and Methods

### 4.1. Fungal Isolation and Purification

The fungus causing die-back disease used in this study was collected from durian plantations in eastern (Chanthaburi and Trat provinces) and southern (Chumphon province) parts of Thailand between April 2019 and October 2021. While collecting disease samples from the durian plantations, we observed the presence of new sawdust outside the bark and noted ambrosia beetles penetrating the durian trees. These observations, combined with recent reports of durian die-back spreading in the region, prompted us to be contacted by durian growers to investigate the disease outbreak. In response, we conducted sampling of infectious tissues from affected plantations. It is important to note that each plantation displayed varying levels of disease severity. This prompted us to collect a significant number of infectious samples, ranging between 10 and 20 samples per location. The GPS coordinates for these sample locations are provided in the Appendix A. To ensure sample integrity, the infected branches were placed in plastic bags and kept in an ice-box for transport to the laboratory. To isolate the fungi, a small piece of infected tissue (approximately 0.5 × 0.5 cm) was excised, and the surface was disinfected with 1.5% sodium hypochlorite for 2–3 min and then rinsed 3 times using distilled sterile water. The tissue was allowed to air and then placed on the surface of potato dextrose agar (PDA) and incubated at 25 °C under a 12 h light/12 h dark photoperiod for 5 days to induce mycelial development. The fungus was purified using the single-spore isolation technique on water agar (WA), and the mycelia were transferred onto potato carrot agar (PCA) and stored at 14 °C for further study.

### 4.2. DNA Extraction and Molecular Identification

For molecular identification of fungal isolates, the total genomic DNA was extracted from fresh mycelial on spezieller nährstoffarmer broth (SNB) for 5 days following the protocol described by Zimand et al. [44] and Rattanakreetakul et al. [23]. The extracted DNA samples were then stored at −20 °C for further analysis. Three genetic regions, the internal transcribed spacer (ITS), translation elongation factor 1 alpha (TEF1-α), and RNA polymerase second-largest subunit (RPB2), were amplified by PCR. The specific primers used for these amplifications are listed in Table 3.

The PCR amplifications were performed in a 25 µL reaction containing 0.96× Taq buffer, 2.4 mM MgCl_2_, 0.48 µM dNTPs, 0.48 µM of each primer, 1 Unit Taq polymerase (Thermo scientific, Co., Ltd., Vilnius, Lithuania), and 2 µL of DNA template (20 ng/µL). The reactions were carried out using PCR thermal cycling (Sensoquest GmbH, Göttingen, Germany) with the following program: pre-denaturation at 94 °C for 3 min, 35 cycles of denaturation at 94 °C for 1 min, annealing (as shown in Table 3) for 1 min, and extension at 72 °C for 1 min, with a final extension step of 72 °C for 10 min. The PCR products were visualized on 1.2% agarose gels using GelStar™ Nucleic acid gel stain and subjected to electrophoresis in 1× TBE buffer at a constant power of 80 volts. The GeneRuler 100 bp Plus DNA Ladder (Thermo Fisher Scientific, Waltham, MA, USA) was used as a size marker for comparison. The gels were photographed under UV radiation with a 365 nm wavelength using a GeneFlash Gel Documentation System (Syngene, Cambridge, UK).

The PCR products were sequenced by the ATCG Co., Ltd., Khlong Nueng, Thailand. Pairwise sequences were analyzed using the EditSeq program of the sequence analysis software version 16.0 (DNAStar Inc., Madison, WI, USA). Basic BLASTn (https://blast.ncbi.nlm.nih.gov/Blast.cgi accessed on 8 January 2023) was performed to compare the nucleotide sequences to the GenBank National Center for Biotechnology Information (NCBI) database, using an E-value cut-off of ≤1 × 10^−5^. The sequences generated in this study were deposited in the GenBank and the accession numbers are shown in Table 3.

### 4.3. Morphological Characteristics

The colony and morphological characteristics under compound microscope were studied as described by *The Genus Fusarium* book [48] and *The Fusarium Laboratory Manual* [30]. The fungal isolates were subcultured on PDA plates and incubated at 25 °C under a 12 h light/12 h dark photoperiod for 5 days to induce mycelial development. A 6 mm plug from the edge of each colony was transferred to the center of the PDA and incubated at 25 °C under a 12 h light/12 h dark photoperiod. The colonies were then inspected for characteristics such as colony type, pigmentation, odor, and colony diameter, which were measured on day 5 after incubation. To examine the conidia, a minimum of 50 conidia were noted using an Olympus CX31 binocular compound microscope at 400× magnification, and the measurements were recorded using Olympus CellSens standard software version 1.16.

### 4.4. Phylogenetic Analysis

The phylogenetic tree was created using the maximum parsimony method with 1000 bootstrap values and the subtree-pruning–regrafting (SPR) algorithm. The evolutionary analyses were conducted in MEGA X [49]. The Bayesian inference (BI) was used to reconstruct the phylogenetic trees using MrBayes version 3.2.7 [50] implemented in the CIPRES cluster (https://www.phylo.org/portal2/home.action (accessed on 8 January 2023)). The nucleotide substitution model was determined by jModelTest v. 2.1.7 [51]. Following Drummond and Rambaut [52], 2,000,000 generations (four chains, four independent runs) were set up. Samples were collected every 1000 generations, and the first 25% of the samples were discarded. The reference isolates used in the analysis are listed in Table 1. The final phylogenetic tree figure was edited using Microsoft Office PowerPoint 2016.

### 4.5. Pathogenicity Test

In this experiment, seventeen representative *Fusarium* isolates were used to inoculate the durian seedlings. The 4-month-old durian seedlings (Mon Thong cultivar) were obtained from a commercial seedling nursery in Chumphon province. Before being transferred to the greenhouse at Kasetsart University, Kamphaeng Saen campus, the seedlings were sprayed with carbendazim (50% W/V SC) at the recommended rate and grown for an additional two months. For the pathogenicity test, a 7 mm in diameter mycelial plug of each fungal isolate was taken from the edge of a 5-day-old fungal pathogen grown on PDA. The mycelial plug was placed onto the durian stem and wrapped with Parafilm (Bemis Packaging, Sheboygan Falls, WI, USA) to prevent desiccation. Pure PDA agar plugs were used as the negative control. After 24 h of incubation, the durian seedlings were placed in a greenhouse under the conditions of 28–34 °C and 71–92% humidity to promote disease development. Twenty-one days after inoculation, the virulence of each isolate was evaluated by the measurement of the lesion length (LL). This experiment design followed a completely randomized design (CRD) with 5 replicates. One-way ANOVA was performed using R software version 3.5.2 [53] with the agricolae package (statistical procedures for agricultural research) [54]. The means of the lesion lengths were compared by Duncan’s new multiple range test. To confirm the causative pathogen of the durian, die-back, Koch’s postulation was used. Re-isolation of the fungal pathogens was performed by collecting four pieces (each 5 mm in diameter) from the disease symptom. The tissues were disinfected with 1.2% sodium hypochlorite for 3 min, and then rinsed three times using sterile distilled water. The tissue was wiped and allowed to air dry, after which it was placed on the PDA and incubated at 25 °C under the conditions mentioned earlier for 5 days to induce mycelial development. Finally, the morphological characteristics of each re-isolated *Fusarium* were studied.

### 4.6. Fusarium–Ambrosia Beetle Association

During the collection of the disease samples from the durian plantations, we noticed the presence of new sawdust outside the bark and observed the ambrosia beetle penetrating into the durian trees. To study this association, we used the adult of *E. similis* code CHA-TW-NOV-1–3 collected from Chanthaburi as a representative ambrosia beetle. The adult of the ambrosia beetle was surface sterilized with 70% ethanol for 1 min and rinsed twice with sterile water, and then transferred onto artificial ambrosia medium [55]. There are two treatments in this experiment: treatment 1 involved transferring an adult ambrosia beetle to a modified medium inoculated with *F. solani* isolate CHA-TW-NOV-1-3/1 and treatment 2 involved transferring one adult ambrosia beetle to the modified medium without *F. solani*. Both treatments were incubated at 25 °C under a 12 h light/12 h dark photoperiod for 48 hr. Five replications were conducted for each experiment. The results were recorded by examining the survival of the ambrosia beetles and dissecting them using scissors and forceps. After dissection, the digestive tract of the beetle was cut and placed on PDA under sterile conditions. This allowed us to perform fungal identification and compare it to the prototype isolates. The care and use of the ambrosia beetles were performed in accordance with the Kasetsart University’s Institutional Animal Care and Use Committee (ID # ACKU65-AGK-042). 

## 5. Conclusions

Our study successfully employed a multilocus sequence analysis (MLSA) and morphological characteristics to effectively identify *Fusarium* species. The pathogenicity assay confirmed that both *F. incarnatum* and *F. solani* play a significant role causing die-back disease in durian. Notably, our study marks the first report of *F. incarnatum* as a causal agent of durian die-back in Thailand. Additionally, we have shed light on the symbiotic relationship between *F. solani* isolate CHA-TW-NOV-1-3/1 and *E. similis* code CHA-TW-NOV-1-3, further emphasizing the potential impact of this association on the health of durian trees. These findings contribute to a deeper understanding of ecological interactions between insects and their associated fungi, highlighting the significance of ambrosia beetle–fungus symbiosis and its possible implications in plant disease transmission.

## Figures and Tables

**Figure 1 plants-12-03045-f001:**
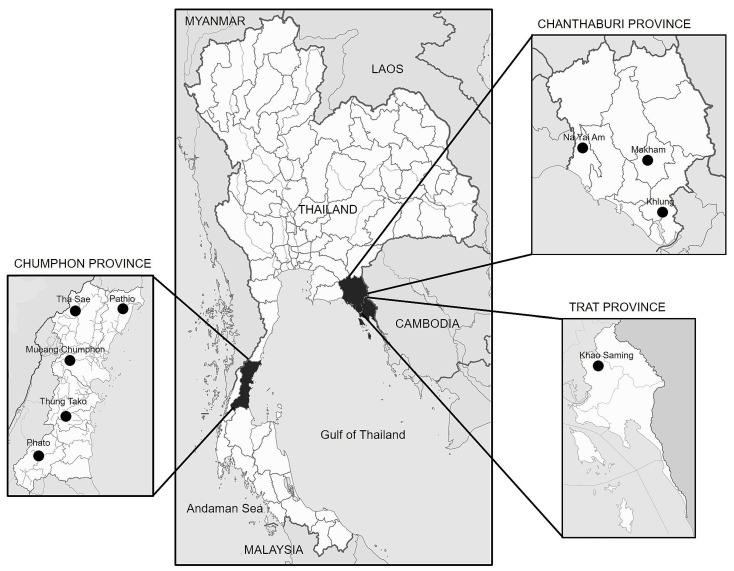
Geographical map showing durian plantation and locations of collected durian die-back samples in the eastern and southern regions of Thailand. Samples were collected between April 2019 and October 2021 (Appendix A).

**Figure 2 plants-12-03045-f002:**
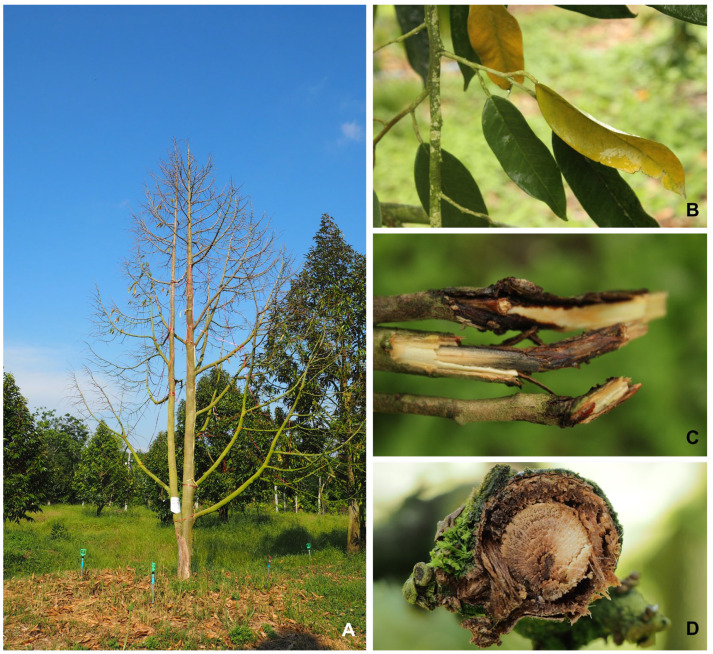
Symptoms of die-back diseases observed on durian trees in a plantation. Die-back on a 5-year-old tree (**A**); yellow leaf at the durian tip (**B**); and transverse cuts showing wood discoloration with brown-to-black lesions in the tissue (**C**,**D**).

**Figure 3 plants-12-03045-f003:**
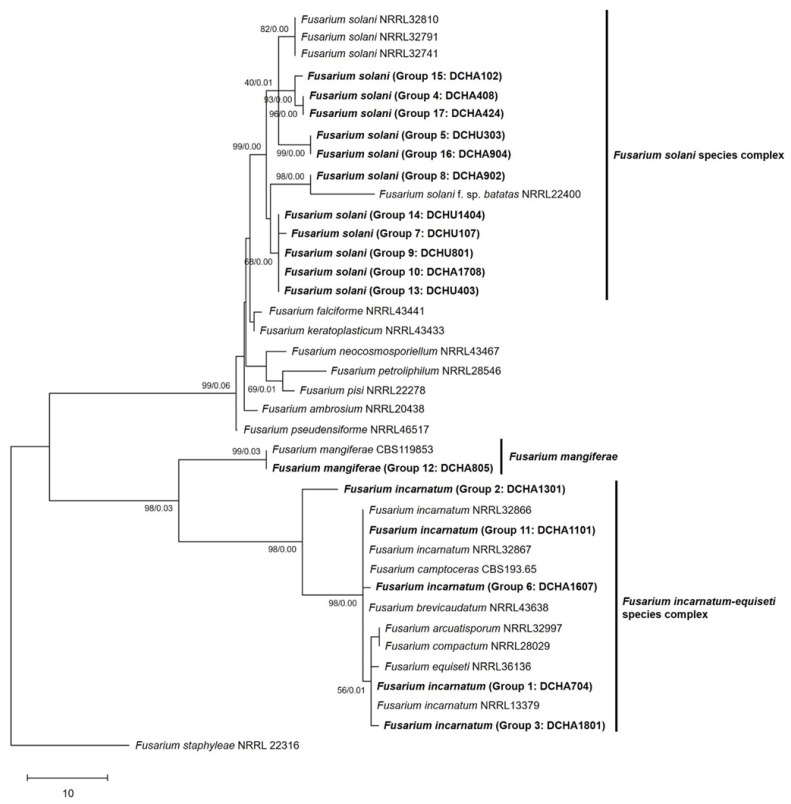
Phylogenetic tree of *Fusarium* isolates, based on maximum parsimony analysis of the ITS region. The percentage of replicate trees in which the associated taxa clustered together in the bootstrap test (1000 replicates) is shown next to the branches. The MP tree was generated using the subtree-pruning–regrafting (SPR) algorithm involving 38 nucleotide sequences. Evolutionary analyses were conducted using MEGA version X. *Fusarium staphyleae* isolate NRRL22316 was used as an outgroup. The present study’s isolates are denoted in bold alphabet.

**Figure 4 plants-12-03045-f004:**
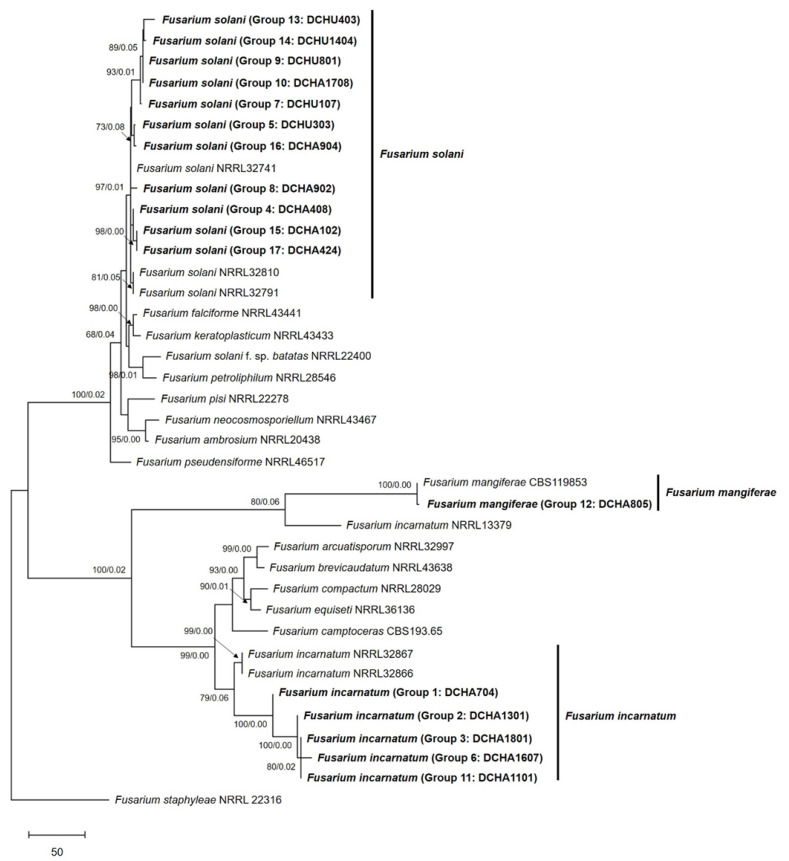
Phylogenetic tree of *Fusarium* isolates, based on maximum parsimony analysis of multilocus sequences, including ITS1-5.8s-ITS2, *TEF1-α,* and *RPB2.* The percentage of replicate trees in which the associated taxa clustered together in the bootstrap test (1000 replicates) is shown next to the branches. The MP tree was generated using the subtree-pruning–regrafting (SPR) algorithm and involved 38 nucleotide sequences. Evolutionary analyses were conducted using MEGA version X. *Fusarium staphyleae* isolate NRRL22316 was used as an outgroup. The present study’s isolates are denoted in bold alphabet.

**Figure 5 plants-12-03045-f005:**
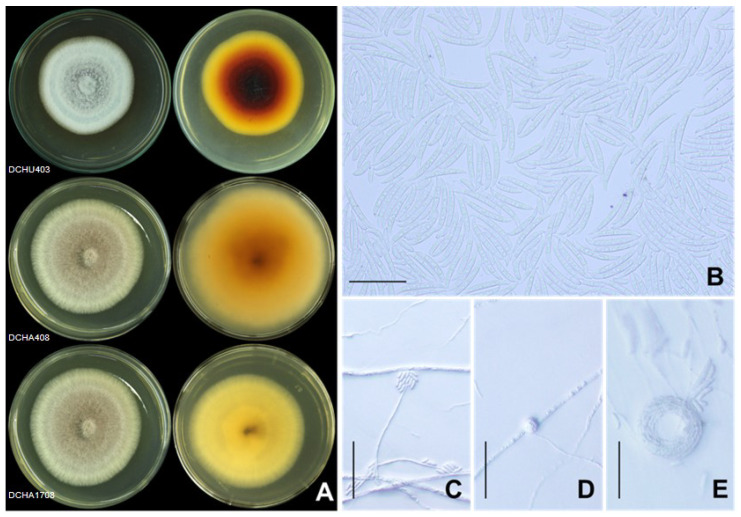
*Fusarium solani*: colony on potato dextrose agar (surface view and reverse view) after 7 d of incubation period at 25 °C (**A**), macroconidia and microconidia (**B**), conidiophore of microconidia (**C**), intercalary chlamydospore (**D**), and hyphal coil (**E**). Scale bars = 50 µm.

**Figure 6 plants-12-03045-f006:**
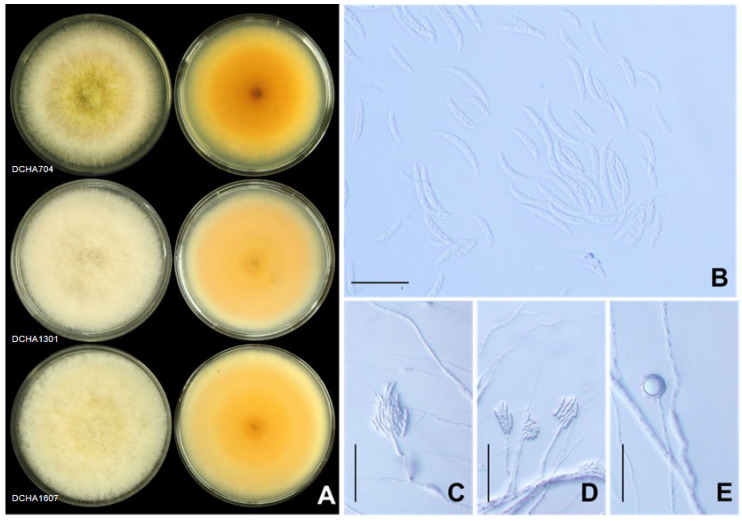
*Fusarium incarnatum*: colony on potato dextrose agar (surface view and reverse view) after 7 d of incubation period at 25 °C (**A**), macroconidia and microconidia (**B**), conidiophore of macroconidia (**C**), conidiophore of microconidia (**D**), and terminal chlamydospore (**E**). Scale bars = 50 µm.

**Figure 7 plants-12-03045-f007:**
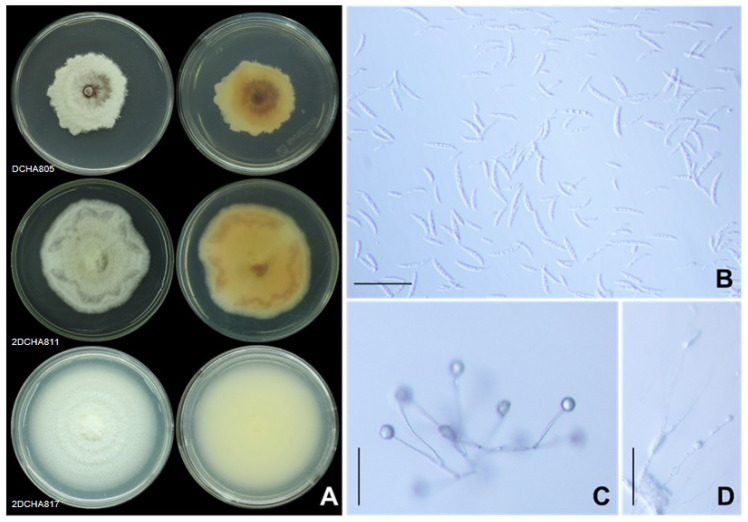
*Fusarium mangiferae*. Colony on potato dextrose agar (surface view and reverse view) after 7 d of incubation period at 25 °C (**A**), macroconidia and microconidia (**B**), conidiophore of microconidia (**C**), and intercalary chlamydospore (**D**). Scale bars = 50 µm.

**Figure 8 plants-12-03045-f008:**
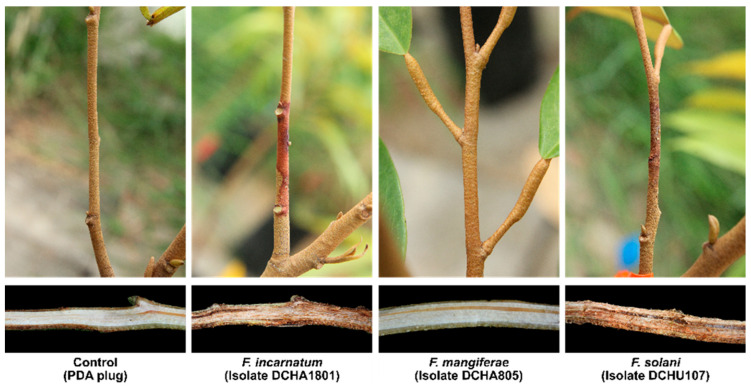
Pathogenicity assays of representative *Fusarium* species using artificial inoculation on durian stems of 6-month-old seedlings. *F. incarnatum* and *F. solani* showed disease symptoms within 21 days after inoculation. In contrast, no disease symptoms were observed with *F. mangiferae*, similar to the control group.

**Figure 9 plants-12-03045-f009:**
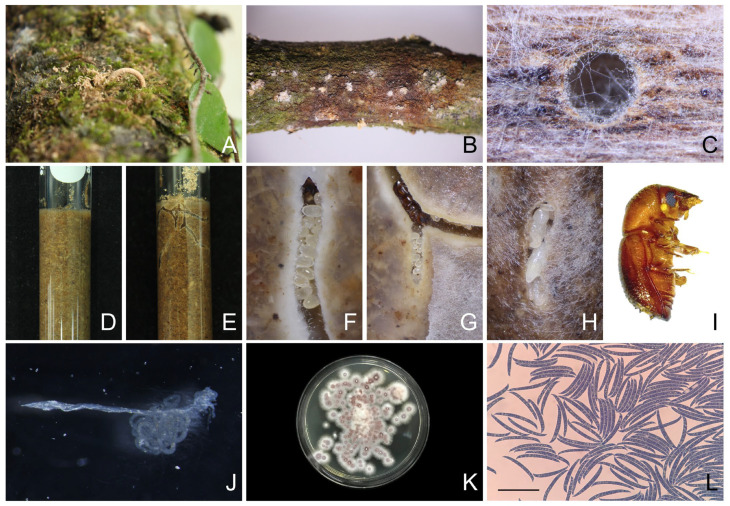
*Fusarium*–ambrosia beetle association in durian plantation and survival of *Euwallacea similis* code CHA-TW-NOV-1-3 on the artificial ambrosia medium with *F. solani* isolate CHA-TW-NOV-1-3/1. Durian sawdust after penetrating on branch in durian plantation (**A**), *Fusarium* fungus covering the disease symptom on a durian branch (**B**), gallery with *Fusarium* fungus (**C**), the artificial ambrosia medium used for the experiment (**D**), galleries created by the ambrosia beetles in the medium inoculated with *F. solani* (**E**), ambrosia eggs in the medium containing *F. solani* (**F**), adult ambrosia beetle with eggs (**G**), pupa of ambrosia beetle (**H**), ambrosia beetle adult (**I**), the digestive track of the ambrosia beetle (**J**), *F. solani* colony detected from digestive track grown on PDA mixed with streptomycin (**K**), and macroconidia similar to *F. solani* CHA-TW-NOV-1-3/1 (40× magnification) (**L**). Scale bars = 50 µm.

**Table 1 plants-12-03045-t001:** Identification and GenBank accessions of *Fusarium* isolates obtained in this study.

Species	Collection ^1^	GenBank Accession Numbers			Identification (GenBank References at 99–100% Similarity)
		ITS	TEF1-α	RPB2	
*Fusarium* Group 1	DCHA704	LC745744.1	LC745761.1	LC745778.1	*F. incarnatum* (NRRL32867, NRRL32866)
*Fusarium* Group 2	DCHA1301	LC745745.1	LC745762.1	LC745779.1	*F. incarnatum* (NRRL32867, NRRL32866)
*Fusarium* Group 3	DCHA1801	LC745746.1	LC745763.1	LC745780.1	*F. incarnatum* (NRRL32867, NRRL32866)
*Fusarium* Group 4	DCHA408	LC745747.1	LC745764.1	LC745781.1	*F. solani* (NRRL32810, NRRL32791)
*Fusarium* Group 5	DCHU303	LC745748.1	LC745765.1	LC745782.1	*F. solani* NRRL32741
*Fusarium* Group 6	DCHA1607	LC745749.1	LC745766.1	LC745783.1	*F. incarnatum* (NRRL32867, NRRL32866)
*Fusarium* Group 7	DCHU107	LC745750.1	LC745767.1	LC745784.1	*F. solani* (NRRL32741)
*Fusarium* Group 8	DCHA902	LC745751.1	LC745768.1	LC745785.1	*F. solani* (NRRL32810, NRRL32791)
*Fusarium* Group 9	DCHU801	LC745752.1	LC745769.1	LC745786.1	*F. solani* (NRRL32741)
*Fusarium* Group 10	DCHA1708	LC745753.1	LC745770.1	LC745787.1	*F. solani* (NRRL32741)
*Fusarium* Group 11	DCHA1101	LC745754.1	LC745771.1	LC745788.1	*F. incarnatum* (NRRL32867, NRRL32866)
*Fusarium* Group 12	DCHA805	LC745755.1	LC745772.1	LC745789.1	*F. mangiferae* (CBS119853)
*Fusarium* Group 13	DCHU403	LC745756.1	LC745773.1	LC745790.1	*F. solani* (NRRL32741)
*Fusarium* Group 14	DCHU1404	LC745757.1	LC745774.1	LC745791.1	*F. solani* (NRRL32741)
*Fusarium* Group 1*5*	DCHA102	LC745758.1	LC745775.1	LC745792.1	*F. solani* (NRRL32810, NRRL32791)
*Fusarium* Group 1*6*	DCHA904	LC745759.1	LC745776.1	LC745793.1	*F. solani* (NRRL32741)
*Fusarium* Group 1*7*	DCHA424	LC745760.1	LC745777.1	LC745794.1	*F. solani* (NRRL32810, NRRL32791)

^1^ NRRL Agricultural Research Service Culture Collection Peoria, Illinois USA; CBS Westerdijk Fungal Biodiversity Institute-KNAW, Utrecht, The Netherlands.

**Table 2 plants-12-03045-t002:** Pathogenicity assay of representative of *Fusarium* groups on 6-month-old durian seedlings.

Group	Species	Representative Isolate	Mean Lesion Size (cm) ± SD
1	*F. incarnatum*	DCHA704	3.35 ± 0.32 e
2	*F. incarnatum*	DCHA1301	2.38 ± 0.34 f
3	*F. incarnatum*	DCHA1801	1.76 ± 0.14 g
4	*F. solani*	DCHA408	3.49 ± 0.27 e
5	*F. solani*	DCHU303	3.94 ± 0.13 d
6	*F. incarnatum*	DCHA1607	2.42 ± 0.11 f
7	*F. solani*	DCHU107	5.05 ± 0.26 c
8	*F. solani*	DCHA902	5.38 ± 0.19 b
9	*F. solani*	DCHU801	5.98 ± 0.08 a
10	*F. solani*	DCHA1708	5.90 ± 0.22 a
11	*F. incarnatum*	DCHA1101	3.98 ± 0.17 d
12	*F. mangiferae*	DCHA805	0.00 ± 0.00 h
13	*F. solani*	DCHU403	3.88 ± 0.17 d
14	*F. solani*	DCHU1404	3.60 ± 0.29 e
15	*F. solani*	DCHA102	4.06 ± 0.07 d
16	*F. solani*	DCHA904	3.94 ± 0.07 d
17	*F. solani*	DCHA424	4.05 ± 0.07 d
F-test			***
C.V. (%)			5.2863
MSE			0.039

*** Values were expressed as mean ± standard error of mean. Values followed by different lowercase superscripts across the rows and different uppercase superscripts along the columns are significantly different at *p* < 0.05. Note: Severity assessment of the isolates based on the length of the die-back on durian at 7 days post-infection. Data are represented as mean ± standard deviation (SD). The mean values followed by different letters indicate significantly different values.

**Table 3 plants-12-03045-t003:** PCR primers for DNA amplification used in this study.

Gene/DNA Region	Primer Details			
Name	Name	Sequence (5′→3′)	Ta (°C)	References
ITS region	ITS5	GGAAGTAAAAGTCGTAACAAGG	56	[45]
	ITS4	TCCTCCGCTTATTGATATGC		
*TEF1-α* gene	EF1-F	ATGGGTAAGGARGACAAGAC	56	[46]
	EF2-R	GGARGTACCAGTSATCATGTT		
*RPB2* gene	fRPB2-7cf	ATGGGYAARCAAGCYATGGG	57.2	[47]
	fRPB2-7cr	CCCATRGCTTGYTTRCCCAT		

## Data Availability

Not applicable.

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
