# Peer review of "Comprehensive Investigation of Die-Back Disease Caused by Fusarium in Durian"

_plants, 2023, doi:10.3390/plants12173045_

Round 1

Reviewer 1 Report

Durian is a famous tropical fruit with extremely high economic value, known as the King of Fruit. The occurrence of die-back disease poses a serious threat to the quality and quantity of durian production in Thailand. The manuscript reported the identification and characterization of the causal agent of durian die-back disease, Fusarium incarnatum, through a combination of colonial characteristics, microscopic morphology, molecular analysis, and plant assays. It was also reported firstly that the ambrosia beetle (Euwallacea similis) plays a significant role in promoting the spread of die-back disease by carrying the pathogens. The topic of the manuscript is from the agricultural practice, and the content of the manuscript will be beneficial to the readers of Plants. The manuscript is well written, but some parts could be improved. Necessary revisions should be made before publication.

Comments and suggestions:

1.        Logically and traditionally, to identify the causal agents of a certain plant disease is firstly to confirm the relatedness between the isolates and the disease by back-inoculation experiments, fulfilling Koch’s Postulates. Why did the authors spend much time and money on characterization of the primary isolates which might contain more redundant and irrelevant organisms? It needs an adequate experimental design (strategy) in identifying the causal agent from a batch of isolates. Please give an explanation about such an experimental design in introduction or in discussion.

2.        Keywords, “transmission” should be involved in the order “Durian; Fusarium; Pathogenicity; Ambrosia beetle; Transmission”.

3.        In 2.1 of the Results, authors should give a general description about the incidence of the disease in certain plantation, the proportion of bacterial and fungal colonies in the primary isolation, and the process in locking the Fusarium as the suspect.

4.        For 2.2, please give the motivation/reason for large-scale gene sequencing without any preliminary screening for such large number of suspects.

5.        For 2.3 and 2.4. Traditionally, the morphological assays and plant assays should go first in pathogen identification.

6.        For 2.5. Plant assays demonstrated that both F. incarnatum and F. solani could effectively infect durian seedlings. Why the authors only emphasize that “This research represents the first report of F. incarnatum as a causal agent of die-back disease in durian in Thailand”? Should F. solani be also in the first report?

7.        For 2.6. Now that the authors repeatedly stressed that “F. incarnatum as a causal agent”, why did not they use F. incarnatum as the test object in Fusarium-ambrosia beetle association study? Any difference in affinity between in F. incarnatum- and F. solani- ambrosia beetle association?

8.        Some subheadings are duplicated in Results and Methods.

9.        The data in Table 3 need an appropriate statistical analysis, e.g., multiple comparisons.

10.     In line 70, “One notable example of this symbiosis is the transmission of Fusarium spp. by the ambrosia beetle Euwallacea sp.”. Is the F. solani-ambrosia beetle association really the Symbiosis?

Author Response

Reviewer1: Comments and suggestions:

  1.   Logically and traditionally, to identify the causal agents of a certain plant disease is firstly to confirm the relatedness between the isolates and the disease by back-inoculation experiments, fulfilling Koch’s Postulates. Why did the authors spend much time and money on characterization of the primary isolates which might contain more redundant and irrelevant organisms? It needs an adequate experimental design (strategy) in identifying the causal agent from a batch of isolates. Please give an explanation about such an experimental design in introduction or in discussion.

Response: We appreciate the reviewer's perspective. It is important to address the concern regarding the allocation of resources and time towards characterizing primary isolates. Respectfully, we hold a differing viewpoint. While the conventional approach involves confirming causal agents through back-inoculation experiments and Koch’s Postulates, our study's focus necessitated a nuanced approach. Upon isolating pathogens from infected tissues, we noted non-Fusarium contaminants evident by distinct colony and spore morphology. To streamline our investigation, we prioritized colonies and spores resembling Fusarium, yielding 86 isolates. Notably, only 16 isolates displayed pathogenicity on 6-month-old durian seedlings. Given our context, employing back-inoculation experiments as per Koch’s Postulates would not have been optimally effective.

  1.   Keywords, “transmission” should be involved in the order “Durian; Fusarium; Pathogenicity; Ambrosia beetle; Transmission”.

Response: Thank you, we agree and made changes as recommended.

  1.   In 2.1 of the Results, authors should give a general description about the incidence of the disease in certain plantation, the proportion of bacterial and fungal colonies in the primary isolation, and the process in locking the Fusarium as the suspect.

Response: Thank you for your feedback. We have incorporated your suggestion by including the following sentences in Section 2.1 of the Results: " The occurrence and prevalence of die-back diseases vary from season to season, depending on weather conditions and the management condition of each plantation. In the regions investigated, the period from April to July corresponds to the monsoon season and warm (26°C-31°C), where the incidence of the disease can reach 70–80% of the total plant."

Regarding the proportion of bacterial and fungal colonies in the primary isolation, and the process in locking the Fusarium as the suspect, we acknowledge the presence of Fusarium as known pathogens affecting Durian trees, and our extensive experience with dieback-related issues. This study's principal aim revolves around delving into the identity of Fusarium at the species level. While performing primary isolations, we did occasionally encounter bacteria and other common fungi. However, for the current study's objectives, we believe that the proportion of bacterial and fungal colonies within the primary isolation is tangential and does not directly contribute to our focal investigation. 

  1.   For 2.2, please give the motivation/reason for large-scale gene sequencing without any preliminary screening for such large number of suspects.

Response: Initially, our approach involved sequencing only the ITS regions as a preliminary screening, aimed at narrowing down our focus specifically to Fusarium. Subsequently, to enhance the precision and accuracy of the identity confirmation, we made the decision to allocate resources towards sequencing two additional well-established markers. This step was undertaken with the goal of obtaining more comprehensive and robust insights into the genetic characteristics of the identified pathogens.

  1.   For 2.3 and 2.4. Traditionally, the morphological assays and plant assays should go first in pathogen identification.

Response: While the traditional approach of commencing with morphological assays and plant assays is indeed valid for numerous studies, our extensive familiarity with Fusarium guided our methodology. In our case, we found it most efficient to initiate the process with colony phenotype assessment, as you suggested. This was then followed by the Multi-Locus Sequence Analysis (MLSA) step before proceeding to the plant assay. This strategy aligns with our in-depth understanding of Fusarium's behavior and variability. As you rightly pointed out, not all Fusarium strains are pathogenic, and it's worth noting that we employed 6-month-old seedlings for our plant assays. This sequence was designed to ensure that we first establish the genetic and phenotypic characteristics of the identified isolates before proceeding to plant-based assessments.

  1.   For 2.5. Plant assays demonstrated that both F. incarnatum and F. solani could effectively infect durian seedlings. Why the authors only emphasize that “This research represents the first report of F. incarnatum as a causal agent of die-back disease in durian in Thailand”? Should F. solani be also in the first report?

Response: We appreciate your observation. While both F. incarnatum and F. solani were shown to infect durian seedlings in our plant assays, the emphasis on "first report" for F. incarnatum is due to its novel identification as a causal agent in durian die-back disease in Thailand. Notably, F. solani had been previously reported by our group in 2020 (Pongpisutta R.;  Rattanakreetakul C.;  Bincader S.;  Chatchaisiri K.; Boonruanfrod P., Detection of fungal pathogen causing durian  713 dieback disease. Khon Kaen Agr. J. 2020, 48 (4), 703-714.  http://10.0.56.120/kaj.2020.65), thus distinguishing it from the context of a "first report." This distinction highlights the original contribution of our study in uncovering F. incarnatum's involvement.

  1.   For 2.6. Now that the authors repeatedly stressed that “F. incarnatum as a causal agent”, why did not they use F. incarnatum as the test object in Fusarium-ambrosia beetle association study? Any difference in affinity between in F. incarnatum- and F. solani- ambrosia beetle association?

Response: We acknowledge your observation. While our focus has been on highlighting F. incarnatum as a primary causal agent, we did conduct various combinations of isolates and ambrosia beetle associations. However, our intention was to underscore F. incarnatum's significance as a newly identified causal agent. The presence of only one combination stems from the availability of F. solani and ambrosia beetles collected from the same plantation (Euwallacea similis code CHA-659 TW-NOV-1-3 and F. solani isolate CHA-TW-NOV-1-3/1). This contrast was aimed at highlighting the unique attributes of F. incarnatum, aligning with the study's emphasis. 

  1.   Some subheadings are duplicated in Results and Methods.

Response: We acknowledge your observation regarding the duplicated subheadings in both the Results and Methods sections. We intentionally duplicated these subheadings to establish a clear alignment between the results and the methods employed. This approach ensures that readers can easily correlate the outcomes with the corresponding methodologies.

  1.   The data in Table 3 need an appropriate statistical analysis, e.g., multiple comparisons.

Response: Thank you for your comment. We want to clarify that we did indeed utilize Duncan's new multiple range test, which is integrated into the agricolae package. This approach is recognized for its effectiveness in conducting multiple comparisons in agricultural research. We have also ensured that this information is correctly reflected in the Materials and Methods section to provide transparency and clarity regarding our statistical analysis methodology. Your feedback is invaluable, and we have taken the necessary steps to appropriately address this concern.

  1. In line 70, “One notable example of this symbiosis is the transmission of Fusarium spp. by the ambrosia beetle Euwallacea sp.”. Is the F. solani-ambrosia beetle association really the Symbiosis?

Response: Based on our comprehensive review of the literature, it appears that various species of ambrosia beetles have indeed formed symbiotic relationships with different species of fungi. Therefore, we maintain the reference to the F. solani-ambrosia beetle interaction as an example of this type of symbiosis in our manuscript. Your input is valuable, and we have ensured that our statement aligns with the established understanding of these ecological relationships.

Reviewer 2 Report

Section 4.1: Fungal Isolation and Purification. Preceding this section, an additional part should be introduced to outline the research framework, encompassing the selection criteria for sampling points at each site, the quantity of plants per point, and supplementary specifics. Thorough descriptions for each site, such as GPS coordinates and other pertinent attributes, should be incorporated.

Section 4.5: Pathogenicity Test. In the Pathogenicity Test section, the quantity of employed pathogenic Fusarium strains should be explicitly indicated. Although this information was referenced in the results section, it should also be incorporated within the Materials and Methods section.

Lines 100-105 in the Results. While commentary on the progression of die-back symptoms development was present within lines 100-105, the Methods section lacks an elucidation of the specific methodology employed for this progression.

Analysis of Fusarium Distribution and Abundance. An essential inclusion involves an analysis of the distribution of Fusarium across the chosen sites and sampling points, along with the relative abundance of distinct Fusarium species obtained from isolation sources.

Table 2 necessitates inclusion solely of sequences derived from the present investigation. To confirm sequence identity, it is imperative to present sequence similarity and the nearest GenBank accession number alongside the obtained sequences within the same row, rather than as separate entries.

Principal Commentary on Section 2.5: Fusarium-Ambrosia Beetle Association. Addressing the study's approach to the Fusarium-ambrosia beetle association, a separate study should be undertaken, carefully considering elements such as study objectives, experimental design, environmental conditions (field or greenhouse), and more. The current study observed by chance the presence of sawdust along with the die-back symptoms and suddenly decided to study the Fusarium-ambrosia beetle association without any previous experimental design to study that association. This unplanned approach raises concerns about the validity of the obtained results.

Fusarium-ambrosia beetle association. The undertaken experiment was elementary, involving the cultivation of ambrosia beetles on artificial medium, with or without F. solani inoculation. However, the mere survival of ambrosia beetles on F. solani-inoculated artificial ambrosia medium does not enough for suggesting a potential symbiotic relationship between the ambrosia beetles and the Fusarium species.

Conclusions Clarification. In light of these considerations, I agree with the conclusion posited in lines 27-28 that this study marks the initial documentation of F. incarnatum as the causal agent of die-back disease in durian of Thailand. Nonetheless, I cannot agree the assertion in lines 28-29 that this study establishes the first account of the link between the ambrosia beetle (Euwallacea similis) and Fusarium, and that ambrosia beetle plays a significant role in promoting the spread of die-back disease caused by Fusarium in durian.

Minor Comments:

- Line 76: "objectives were threefold: (1)" should be revised to "objectives were: (1)"

- Line 80-81: "this study is twofold, firstly," should be rephrased to eliminate "twofold"

- The misuse of "two or threefold" in these sentences requires correction.

Minor editing of English language required

Author Response

Reviewer2: Comments and suggestions:

Section 4.1: Fungal Isolation and Purification. Preceding this section, an additional part should be introduced to outline the research framework, encompassing the selection criteria for sampling points at each site, the quantity of plants per point, and supplementary specifics. Thorough descriptions for each site, such as GPS coordinates and other pertinent attributes, should be incorporated.

Response: we have incorporated the following sentences preceding the section:

While collecting disease samples from the durian plantations, we observed the presence of new sawdust outside the bark and noted ambrosia beetles penetrating the durian trees. These observations, combined with recent reports of durian dieback spreading in the region, prompted us to be contacted by durian growers to investigate the disease outbreak. In response, we conducted sampling of infectious tissues from affected plantations. It's important to note that each plantation displayed varying levels of disease severity. This prompted us to collect a significant number of infectious samples, ranging between 10 to 20 samples per location. The GPS coordinates for these sample locations are provided in a supplemental Table S1.

Section 4.5: Pathogenicity Test. In the Pathogenicity Test section, the quantity of employed pathogenic Fusarium strains should be explicitly indicated. Although this information was referenced in the results section, it should also be incorporated within the Materials and Methods section.

Response: Per you advised, we have included the following sentence in the first sentence of the Pathogenicity Test section within the Materials and Methods section:

In this experiment, seventeen representative Fusarium isolates were used to inoculate durian seedlings.

Lines 100-105 in the Results. While commentary on the progression of die-back symptoms development was present within lines 100-105, the Methods section lacks an elucidation of the specific methodology employed for this progression.

Response: We have included the following information within the Methods section:

To ensure sample integrity, the infected branches were placed in plastic bags and kept in an ice-box for transport to the laboratory.

Analysis of Fusarium Distribution and Abundance. An essential inclusion involves an analysis of the distribution of Fusarium across the chosen sites and sampling points, along with the relative abundance of distinct Fusarium species obtained from isolation sources.

Response: We added the following sentences in the Results section (2.2). Intriguingly, an analysis of the distribution and relative abundance of Fusarium across the selected sites and sampling points revealed that F. incarnatum was the predominant species (53/86), followed by F. solani (30/86) and F. mangiferae (3/86), as documented in Tables S1 and S2.

Table 2 necessitates inclusion solely of sequences derived from the present investigation. To confirm sequence identity, it is imperative to present sequence similarity and the nearest GenBank accession number alongside the obtained sequences within the same row, rather than as separate entries.

Response: While we believe the phylogenetic analysis offers similar information, we understand your concern. To address this, we have taken your feedback into consideration. In the Results section, we have added the following sentence to provide clarity on sequence comparison:

Basic BLASTn (https://blast.ncbi.nlm.nih.gov/Blast.cgi) was performed to compare the nucleotide sequences to the GenBank National Center for Biotechnology Information (NCBI) database using an E-value cut-off of ≤1e-5.

We appreciate your understanding that adding another column to Table 2 might lead to overcrowding. Your feedback is invaluable in refining the information within the manuscript.

Principal Commentary on Section 2.5: Fusarium-Ambrosia Beetle Association. Addressing the study's approach to the Fusarium-ambrosia beetle association, a separate study should be undertaken, carefully considering elements such as study objectives, experimental design, environmental conditions (field or greenhouse), and more. The current study observed by chance the presence of sawdust along with the die-back symptoms and suddenly decided to study the Fusarium-ambrosia beetle association without any previous experimental design to study that association. This unplanned approach raises concerns about the validity of the obtained results.

Response: We want to clarify that our decision to explore this association was not solely based on the presence of sawdust, but rather a combination of factors. We observed the presence of sawdust in close proximity to die-back symptoms, which prompted our interest. Additionally, existing literature has provided insights into the association between fungi and ambrosia beetles. To address these points, we have included relevant information in the introduction section, explaining the rationale behind our investigation into the Fusarium-ambrosia beetle association (Lines 62-75).

Fusarium-ambrosia beetle association. The undertaken experiment was elementary, involving the cultivation of ambrosia beetles on artificial medium, with or without F. solani inoculation. However, the mere survival of ambrosia beetles on F. solani-inoculated artificial ambrosia medium does not enough for suggesting a potential symbiotic relationship between the ambrosia beetles and the Fusarium species.

Response: we agree that mere survival might not be enough to suggest a potential symbiotic relationship. However, we would like to point out that the approach we employed aligns with established methods used by entomologists to assess the association between beetles and fungi. We have referenced literature (references 12, 21, 31, 36, 43, and 54) that supports the utilization of similar methodologies for studying these relationships. While this experiment might be elementary, it serves as an initial step to explore the potential interaction between ambrosia beetles and Fusarium species. We appreciate your perspective and assure you that we have taken into account the standard practices within the field. 

Conclusions Clarification. In light of these considerations, I agree with the conclusion posited in lines 27-28 that this study marks the initial documentation of F. incarnatum as the causal agent of die-back disease in durian of Thailand. Nonetheless, I cannot agree the assertion in lines 28-29 that this study establishes the first account of the link between the ambrosia beetle (Euwallacea similis) and Fusarium, and that ambrosia beetle plays a significant role in promoting the spread of die-back disease caused by Fusarium in durian.

Response: In light of these considerations, while there have been reports suggesting the association of other beetle species with Fusarium, this study presents the first documented evidence of the association between Euwallacea similis and F. solani at the species level. We acknowledge that you find the assertion in lines 28-29 questionable. Therefore, we have revised our conclusion to state, Additionally, this study uncovers the association of ambrosia beetles and F. solani, highligthing the potential involvement of E. similis in facilitating the spread of die-back disease caused by Fusarium in durian.

Minor Comments:

- Line 76: "objectives were threefold: (1)" should be revised to "objectives were: (1)"

- Line 80-81: "this study is twofold, firstly," should be rephrased to eliminate "twofold"

- The misuse of "two or threefold" in these sentences requires correction.

Response: We made all these changes as suggested.

Reviewer 3 Report

The research article titled "Comprehensive Investigation of Die-Back Disease Caused by Fusarium in Durian" presents intriguing and innovative insights into the dieback disease. The study's findings are thoughtfully portrayed and effectively presented. Based on the quality of its content and presentation, I recommend the publication of this article in the upcoming issue of the 'Plant ' journal.

Author Response

Reviewer 3: Comments and suggestions:

Comments and Suggestions for Authors

The research article titled "Comprehensive Investigation of Die-Back Disease Caused by Fusarium in Durian" presents intriguing and innovative insights into the dieback disease. The study's findings are thoughtfully portrayed and effectively presented. Based on the quality of its content and presentation, I recommend the publication of this article in the upcoming issue of the 'Plant ' journal.

Response: We appreciate the time and effort that you have dedicated to providing your valuable feedback on our manuscript. Moreover, we also very much recognize for your support.

Round 2

Reviewer 1 Report

The authors have carefully revised the manuscript, which has significantly improved.

Please add "Durian" as the keyword.

Author Response

Reviewer1: Comments and suggestions:

  1.   The authors have carefully revised the manuscript, which has significantly improved.

Please add "Durian" as the keyword.

Response: We appreciate the reviewer's suggestion, we added “Durian” into a keyword section.

*********************************************************************

Response: We made all these changes as suggested.

Thank you very much for giving us the opportunity to revise our manuscript. I would like to thank the reviewers for their valuable comments.

Reviewer 2 Report

The manuscript entitled "Comprehensive Investigation of Die-Back Disease Caused by Fusarium in Durian” investigated die-back disease in durian plantations located in Thailand through a combination of colony characteristics, microscopic morphology, and multilocus sequence analysis (MLSA) of internal ITS region, 1-α (TEF1-α) gene, and RPB2 sequences. The current study tried to report the relationship between the ambrosia beetle (Euwallacea similis), Fusarium, and die-back disease.

The manuscript was improved after authors revision and they addressed most of my comments. However, Table 2 still needs more modification, I did not ask to add more columns. Table 2 should only include the sequences which were obtained from the present study. Sequence similarity and the nearest GenBank accession number should be provided to confirm the identification at the same row of the obtained sequences, not as were provided as separated rows. Relative abundance of different Fusarium species within the selected sites and points should be presented as a percentages (%).

Author Response

Reviewer2: Comments and suggestions:

The manuscript entitled "Comprehensive Investigation of Die-Back Disease Caused by Fusarium in Durian” investigated die-back disease in durian plantations located in Thailand through a combination of colony characteristics, microscopic morphology, and multilocus sequence analysis (MLSA) of internal ITS region, 1-α (TEF1-α) gene, and RPB2 sequences. The current study tried to report the relationship between the ambrosia beetle (Euwallacea similis), Fusarium, and die-back disease.

The manuscript was improved after authors revision and they addressed most of my comments. However, Table 2 still needs more modification, I did not ask to add more columns. Table 2 should only include the sequences which were obtained from the present study. Sequence similarity and the nearest GenBank accession number should be provided to confirm the identification at the same row of the obtained sequences, not as were provided as separated rows. Relative abundance of different Fusarium species within the selected sites and points should be presented as a percentages (%).

Response: We concur and apologize for this oversight.

In Table 2, we removed the nearest GenBank entries from the separate rows and, as advised, incorporated them into the modified last column.

"Regarding the 'Relative abundance of different Fusarium species within the selected sites and points should be presented as percentages (%)”, we have made the appropriate correction. It is now read “Intriguingly, an analysis of the distribution and relative abundance of Fusarium across the selected sites and sampling points revealed that F. incarnatum was the predominant species (61.63%), followed by F. solani (34.88%) and F. mangiferae (3.49%), as documented in Tables S1 and S2.””

**********************************************************************************

Response: We made all these changes as suggested.

Thank you very much for giving us the opportunity to revise our manuscript. I would like to thank the reviewers for their valuable comments.
